# Established and Novel Risk Factors for 30-Day Readmission Following Total Knee Arthroplasty: A Modified Delphi and Focus Group Study to Identify Clinically Important Predictors

**DOI:** 10.3390/jcm12030747

**Published:** 2023-01-17

**Authors:** Daniel Gould, Michelle Dowsey, Tim Spelman, James Bailey, Samantha Bunzli, Siddharth Rele, Peter Choong

**Affiliations:** 1Department of Surgery, University of Melbourne, St. Vincent’s Hospital Melbourne, Melbourne, VIC 3065, Australia; 2Department of Orthopaedics, St. Vincent’s Hospital Melbourne, Melbourne, VIC 3065, Australia; 3School of Computing and Information Systems, University of Melbourne, Melbourne, VIC 3010, Australia; 4Nathan Campus, School of Health Sciences and Social Work, Griffith University, Brisbane, QLD 4111, Australia; 5Physiotherapy Department, Royal Brisbane and Women’s Hospital, Brisbane, QLD 4029, Australia

**Keywords:** knee, arthroplasty, risk factors, Delphi survey, focus group, readmission, clinical

## Abstract

Thirty-day readmission following total knee arthroplasty (TKA) is an important outcome influencing the quality of patient care and health system efficiency. The aims of this study were (1) to ascertain the clinical importance of established risk factors for 30-day readmission risk and give clinicians the opportunity to suggest and discuss novel risk factors and (2) to evaluate consensus on the importance of these risk factors. This study was conducted in two stages: a modified Delphi survey followed by a focus group. Orthopaedic surgeons and anaesthetists involved in the care of TKA patients completed an anonymous survey to judge the clinical importance of risk factors selected from a systematic review and meta-analysis and to suggest other clinically meaningful risk factors, which were then discussed in a focus group designed using elements of nominal group technique. Eleven risk factors received a majority (≥50%) vote of high importance in the Delphi survey overall, and six risk factors received a majority vote of high importance in the focus group overall. Lack of consensus highlighted the fact that this is a highly complex problem which is challenging to predict and which depends heavily on risk factors which may be open to interpretation, difficult to capture, and dependent upon personal clinical experience, which must be tailored to the individual patient.

## 1. Introduction

Hospital readmission is a widely used metric to assess the quality of care in several countries [1]. In recent years, efforts to improve outcomes and reduce complications, including readmission rate, following TKA have led to a surge in innovation in terms of alignment [2], prosthetic manufacture [3], and assistance in the surgical procedure with robotics [4] and other devices [5].

Systematic reviews and meta-analyses of risk factors for 30-day readmission following TKA demonstrated that there is a broad range of risk factors, many of which have a weak effect on readmission risk [6,7]. When many potential risk factors exist for an important clinical problem with few strong risk factors, insight from clinicians regarding the clinical importance of these risk factors is critically important [8]. For this purpose, the Delphi survey technique is a robust method of obtaining this insight on clinical importance and for evaluating consensus on a specific real-world issue [9]. It is particularly well-suited to answering questions of prognostication [10] because experts anonymously appraise risk factors in terms of their importance as predictors of a clinical outcome [8,11,12,13,14]. Such critical appraisal of available information on a curated set of risk factors identified through a systematic literature review strengthens the clinical applicability of these risk factors. This is useful in risk evaluation for clinical outcomes because it highlights which patient characteristics should be factored into an individual patient’s risk profile, even if it is not a strong independent risk factor [15]. The Delphi technique also facilitates recommendations of risk factors not identified in prior literature [8]. Novel risk factors suggested by clinicians can then be discussed in a focus group which facilitates critical appraisal of suggested risk factors and provides deep insight into their clinical importance [16]. A powerful aspect of this format is that it enables clinicians to discuss the reasons why certain risk factors may or may not be clinically relevant [17]. This has the potential to go beyond the appraisal of individual risk factors to provide insight into the complexities of the clinical problem more broadly [18].

The purpose of this study was twofold and involved engaging with clinicians involved in the care of TKA patients. The first aim was to ascertain the clinical importance of established risk factors in the evaluation of 30-day readmission risk and give clinicians the opportunity to suggest and discuss novel risk factors. The second aim was to evaluate consensus on the importance of risk factors identified from the literature as well as novel risk factors identified by study participants, considering this is a complex clinical problem for which risk evaluation is challenging, and views may vary widely depending on personal clinical experience and perception.

## 2. Materials and Methods

This mixed methods study was conducted in 2 stages. The first stage involved a modified Delphi survey [10], and the second stage involved a focus group conducted using elements of the nominal group technique [19].

### 2.1. Stage 1: Survey

#### 2.1.1. Study Design

A modified Delphi approach [10,20] was used to develop a survey (Qualtrics version 01.22, Provo, UT, USA) based on systematic review and meta-analysis findings on patient-related risk factors for 30-day readmission following TKA [6]. Only predictors with high or moderate quality of evidence as determined through a modified Grading of Recommendations Assessment, Development and Evaluation (GRADE) approach were selected [6,21]. This decision was made to exclude variables for which there was poor quality evidence.

In total, there were 50 risk factors included in the survey. Participants were asked to indicate the importance of each risk factor when considering a patient’s overall risk of readmission within 30 days following primary TKA, with a choice of 3 options: low importance, moderate importance, and high importance. Free text fields were also generated to enable respondents to suggest risk factors that were not already listed. Responses were anonymised, but respondents were asked to indicate their designation and amount of experience with TKA patients. Participants could not see each other’s responses. A copy of the Qualtrics survey is available in the Appendix A. This also contains a summary of the systematic review evidence for each risk factor, which was included to better inform clinicians’ votes.

#### 2.1.2. Recruitment

The survey was pilot tested with a final-year MD student and an MD-PhD candidate (author SR).

Following changes made in accordance with feedback obtained through pilot testing, the survey was sent to all consultant orthopaedic surgeons, consultant anaesthetists, orthopaedic registrars, and orthopaedic residents at 11 regional and metropolitan centres across Victoria, Australia. In Australia, consultants are senior clinicians who have completed their training, registrars are clinicians undergoing speciality training, and residents are junior doctors yet to commence speciality training. These groups of clinicians were selected because they are the most heavily involved in the care of TKA patients. Clinicians of varied levels of seniority were recruited because they may offer valuable insight based on their level of clinical experience compared to the recency of their formal medical school education [22]. Anaesthetists were included to provide a different clinical perspective than that of orthopaedic surgeons because diverse stakeholder input into Delphi studies can offer valuable insight [11]. The perspective of anaesthetists is particularly pertinent to the appraisal of 30-day readmission risk factors in TKA patients, given their unique expertise in perioperative medicine and risk evaluation [23]. A minimum of 30 participants was sought per recommendations from prior literature [14]. The response rate was calculated based on the number of clinicians who were sent the survey and the number of responses received.

#### 2.1.3. Analysis and Software

Data were exported from Qualtrics and imported to Microsoft Excel (version 16.0, Redmond, WA, USA) for data preparation. The primary analysis involved calculating the proportion of votes in each category: low, moderate, and high importance. A simple majority (≥50%) was used to determine the importance of each risk factor. Risk factors were then ranked from most to least important according to the proportion of votes in the category with the majority of votes.

Secondarily, the consensus was evaluated. Krippendorff’s alpha [24,25] was used because the data were ordinal and involved multiple observers. This calculation was carried out in R statistical software (version 4.1.1, Vienna, Austria) [26] using the implementation provided by Zapf et al. [24]. An alpha value of 0.6 was considered the minimum requirement for consensus, with values higher than 0.6 indicating stronger consensus [27].

### 2.2. Stage 2: Focus Group

#### 2.2.1. Study Design

Elements of the nominal group technique [19,28] were utilised to design a focus group session in which all participants would be given equal opportunity to contribute to the discussion in a systematic manner around the clinical importance of each risk factor. This granted each participant an equal opportunity to consider the opinions of others and voice their own opinion. The steps of a strict nominal group technique discussion are as follows: silent idea generation, round robin, clarification, and voting [19]. Participants in the present study were voting on a pre-determined list of risk factors suggested in the Delphi survey stage rather than suggesting risk factors independently; hence the silent generation portion was replaced by the initial vote on these previously-suggested risk factors. This format is distinct from the open-ended discussions characteristic of other focus group designs. The rationale for the decision to draw on nominal group technique for this focus group was twofold: (1) to mitigate the power imbalance resulting from having senior clinicians and junior clinicians in the focus group, and (2) to maintain focus of the discussion on the risk factor being discussed, rather than focusing on interaction between, or personal experience of, dominant group members. Specifically, the study coordinator (DG) facilitated discussion in the manner depicted in Figure 1.

The focus group was carried out via online video conference due to limitations on in-person gatherings imposed at the time due to the COVID-19 pandemic. Poll Everywhere was utilised [29] to facilitate voting on each risk factor. This enabled participants to see the live results of their votes and change their individual responses in real time if they were persuaded by the rationale provided by other participants whose views differed from their own.

Participants addressed the study coordinator (DG) rather than speaking directly to one another. Discussion after the initial vote enabled group members to clarify, challenge, and justify their interpretation and appraisal of each risk factor. The duration of this focus group discussion was 1 h. After a brief introduction, during which the study coordinator outlined how the discussion was to be facilitated and what was expected of participants, each risk factor was discussed in turn until no group members had any further contributions, and the final vote was counted. Each group member recorded both their initial and final vote on a worksheet to document changes in responses that occurred through discussion (see Appendix A for a blank version of this worksheet).

The focus group design was pilot tested with 8 PhD candidates (including author SR).

#### 2.2.2. Recruitment

The heads of the departments of orthopaedics and anaesthetics at a metropolitan hospital sent a recruitment email to their staff inviting participation. This hospital is a tertiary referral centre for TKA in Victoria and serves a large and diverse population [30]. The target for recruitment comprised 2 representatives from each of the following groups: consultant anaesthetists, consultant orthopaedic surgeons, orthopaedic registrars, and orthopaedic residents. Ideally, 1 male and 1 female participant would be sought from each representative group. Verbal consent was recorded for each participant prior to the focus group session. Those for whom verbal consent was obtained were sent a Doodle Poll [31] to select a date and time at which at least 5 participants could attend, as the target sample size was between 5 and 9 participants (inclusive) [32].

#### 2.2.3. Analysis and Software

Risk factor importance and consensus were evaluated in the same manner as the Delphi stage of this study.

To gain a better understanding of the clinicians’ rationale for their risk factor evaluations, the audio recording of the focus group discussion was transcribed verbatim. The portion of the transcript for which participants changed their votes following discussion was analysed thematically according to the 6-step process outlined by Braun and Clarke [33]: (a) familiarization; (b) initial coding; (c) identifying themes; (d) reviewing themes; (e) defining themes; and (f) reporting. DG identified codes and, together with SB, grouped the codes into categories and identified emergent themes. These themes were discussed between SB and DG and subsequently defined in line with the coding framework. Themes were presented in narrative form with supporting quotes from focus group participants to illustrate that the interpretations were grounded in the participants’ perspectives.

For the thematic analysis of the focus group, the transcript of the audio recording was imported into the NVivo data management package (version 12.0, Burlington, VT, USA).

## 3. Results

### 3.1. Stage 1: Survey

#### 3.1.1. Recruitment

There were 39 participants who commenced the survey, but three of these did not answer any questions. Two consented to participate but did not continue any further, and the third did not complete the ‘Do you consent to participate?’ question. Of the remaining 36 respondents, there were 30 participants who provided a vote on all risk factors.

The median time taken to complete the survey for all 36 participants was 10 min (interquartile range = 7–14 min).

Response rates for orthopaedic surgeons and anaesthetists were difficult to calculate due to a lack of response from some site representatives and difficulty ascertaining the number of available clinicians at each site. Approximately 107 consultant anaesthetists were sent the survey, yielding a response rate of 20/107 = 18.69%. Approximately 77 orthopaedic clinicians (consultants, registrars, and residents) were sent the survey, yielding a response rate of 16/77 = 20.78%. However, six of the 11 site representatives did not respond to the recruitment email; therefore, the calculated response rates were likely an underestimate because the numerator (i.e., number of respondents) is possibly only provided for five of 11 sites, while the denominator represents the approximate number of respondents at all 11 sites. Therefore, the true response rate is likely somewhat closer to that of prior published Delphi survey studies on prognostic factors [8,14]. Table 1 depicts the characteristics of the survey participants.

Not included in Table 1 were the details of the three participants who commenced the survey but did not respond to any questions. One provided no details. The remaining two were consultant anaesthetists, one with 2 years of experience and a caseload of 10 TKAs per year, the other with 19 years of experience and a caseload of 10 TKAs per year.

#### 3.1.2. High-Importance Predictors

Figure 2 depicts a bar chart comprising the risk factors included in the Delphi survey, in descending order of the proportion of votes in the high-importance category. This figure contains all 36 survey participants’ responses, with missing values included for risk factors where participants did not provide a vote.

Appendix A contains the numbers used to produce Figure 2. This table also indicates which risk factors received a majority (≥50%) high-importance vote despite a lack of systematic review evidence. These risk factors were: return to theatre, ICU or HDU admission, and preoperative patient-reported pain level. The table also indicates which risk factors did not receive a majority vote of high importance, despite systematic review evidence demonstrating a correlation with readmission: anaemia, low socioeconomic status, chronic kidney disease, coagulopathy, depression, age, arrhythmia, history of cancer, peripheral vascular disease, hypertension, race, and male sex.

Table 2 depicts the Delphi survey risk factors with a majority vote (≥50%) of high importance.

An important consideration for the interpretation of these findings for the anaesthetists is captured in a comment made by a Delphi survey participant (consultant anaesthetist, 30 years of experience, caseload 15 years): “as an anaesthetist, I am not aware of when a patient is readmitted and so have little to base my judgement on”.

#### 3.1.3. Consensus

The values for Krippendorf’s alpha were low overall and in each subgroup (consultant anaesthetists, consultant orthopaedic surgeons, orthopaedic registrars, and all orthopaedic clinicians as a combined group). The highest value was for orthopaedic registrars: 0.27 (95% confidence interval (CI) = 0.05–0.47). This is substantially lower than the minimum alpha value considered as evidence for consensus (0.6).

Appendix A depicts alpha values for the Delphi participants overall, as well as each subgroup.

### 3.2. Stage 2: Focus Group

#### 3.2.1. Risk Factor Selection

Risk factors suggested by Delphi survey participants are listed in Appendix A. To reduce the number of risk factors to a manageable level for discussion in the focus group, related risk factors were grouped into categories. Twenty-two risk factors were discussed in the one-hour session.

#### 3.2.2. Recruitment

Ten clinicians provided consent to participate. Three were consultant anaesthetists—one female and two male. Four were consultant orthopaedic surgeons—one female and three male. Three were orthopaedic registrars—two female and one male.

Six of the 10 clinicians who provided consent participated in the scheduled focus group session, meeting the targets for sample size and diversity. Characteristics of the six participants are presented in Table 1.

#### 3.2.3. High-Importance Risk Factors Identified through Focus Group Discussion

Figure 3 depicts a bar chart comprising the risk factors voted on in the focus group following a discussion, in descending order of the proportion of votes in the high-importance category.

Appendix A contains vote counts for each focus group risk factor. This includes an indication of how votes changed between the initial and final votes. Appendix A depicts a bar chart comprising the risk factors vote counts on the initial vote prior to discussion.

The findings pertaining to high-importance variables (≥50% majority vote) according to the final vote after discussion are depicted in Table 3 below. The high-importance risk factors in subgroups with only two participants were those for which at least one participant ranked it as high importance. Appendix A contains these findings for the initial vote.

#### 3.2.4. Thematic Analysis of Focus Group Transcript

Those risk factors for which participants changed their votes are included in Appendix A, along with verbatim excerpts from the transcript.

This portion of the transcript, for which there was a change in votes through discussion, was analysed thematically. The participants’ responses were coded, and these codes were grouped into five categories. Two themes emerged from this process: uncertainty and adaptability. The coding tree, including codes, code categories, and emergent themes, is included in Appendix A.

#### 3.2.5. Uncertainty Theme: Clinicians Are Faced with Uncertainty in the Definition, Relevance, Potency, and Generalisability of Risk Factors for 30-Day Readmission Following TKA

Uncertainty characterises multiple facets of 30-day readmission risk factor evaluation. Ambiguity in the definition of various risk factors led to different interpretations of the risk factor and, therefore, different appraisals of its importance, with one participant (consultant anaesthetist) saying the following regarding the risk factor ‘surgical factors (prolonged/complex/difficult, surgical misadventure)’: “that’s my reading of a statement that has mixed things in it. Surgical misadventure is dramatically different from just a difficult citing of the prosthesis”. Compounding this is the difficulty clinicians face in ranking the relative importance of many potential risk factors in relation to one another, as well as determining which of an individual patient’s risk factors is most impactful for their personal risk profile. One participant (consultant orthopaedic surgeon) said the following about the risk factor ‘poor understanding of disease and post-op course’: “it is not that it’s not important, but … I think that it’s probably a fairly low importance in terms of causes of readmission. There are other things more likely”. Clinicians must also consider whether the risk factor in question is generalisable to other settings or if it is specific to the nuances of the policies and processes in place at a particular institution or setting. An orthopaedic registrar suggested the following about the risk factor ‘poor compliance with rehabilitation’: “it may not be a St Vincent’s risk factor as much, because that may be less the protocol, but certainly a wider Australia that’s how I’d read it”.

Adaptability theme: faced with uncertainty around the specific aspects of the risk profile of patients likely to be readmitted within 30 days of their TKA surgery, clinicians seek to adapt their approach to tailoring their risk evaluation and subsequent management plan to the needs and circumstances of the individual patient.

Given the uncertainty around risk factors for 30-day readmission following TKA, clinicians aim to strike a balance between utilising personal clinical experience and objective scientific evidence. Objective evidence may be scarce, but clinicians draw on it when it is available in order to better inform their decisions regarding risk characterisation and mitigation, as indicated by an orthopaedic registrar commenting on ‘*resilience*’ as a potential risk factor: “there is a […] brief resiliency scale […] which is used in research as a way of looking at outcome in joint arthroplasty, and it’s relatively highly associated with poorer outcomes. So then I had it as a risk factor, therefore, for those poor outcomes are often readmissions as well”. Armed with objective evidence as well as intuition honed through clinical experience, clinicians evaluate the complex risk profile of their patients and attempt to personalise care to suit their needs such that their specific risk factors are addressed. The same risk factor has varying degrees of severity in different patients, and clinicians must personalise care to the individual patient’s needs rather than take a generic approach where it is assumed the same management strategy will be effective for a given risk factor in different patients. An orthopaedic registrar captured this when discussing the ‘*multi-joint disease*’ risk factor: “I think it depends on how you manage those patients while they’re inpatients, and educate as well”.

#### 3.2.6. Consensus

As with the Delphi stage of this study, there was a lack of consensus in the focus group, with none of the Krippendorf’s alpha values reaching the threshold of 0.6. The consensus calculations for the focus group are available in Appendix A. The main findings are summarised here for both the initial vote and, following discussion, the final vote. The highest degree of initial vote consensus was achieved among the consultant anaesthetists (0.42 (95% CI = −0.04–0.75)) and orthopaedic registrars (0.35 (95% CI = −0.17–0.76)). At the final vote, the highest consensus occurred amongst consultant orthopaedic surgeons (0.26 (95% CI = −0.22–0.68)) and orthopaedic registrars (0.28 (95% CI = −0.27–0.75)). For consultant orthopaedic surgeons, this change represented an improvement in the level of consensus through focus group discussion. A similar finding was observed for the combined orthopaedic group (consultants and registrars). However, overall focus group consensus decreased slightly from 0.18 (95% CI = <−0.01–0.39) to 0.1633 (95% CI = −0.03–0.41).

## 4. Discussion

In this study, clinicians involved in the care of TKA patients were engaged in utilising their clinical training and experience in the appraisal and suggestion of risk factors for 30-day readmission. Eleven risk factors received a majority (≥50%) vote of high importance in the Delphi survey overall: return to theatre, any in-hospital complication, intensive care unit or high-dependency unit admission, dependent functional status, dementia, preoperative patient-reported pain level, liver disease, Charlson Comorbidity Index, substance abuse, increasing number of previous admissions, and congestive heart failure. Six risk factors received a majority vote of high importance in the focus group overall: inadequate pain management at discharge, pain catastrophizing or analgesia intolerance/catastrophic pain, high risk of infection, transplant recipient, the threshold for readmission, and surgical factors (prolonged/difficult surgery). None of the Krippendorff’s alpha statistics reached the threshold value (0.6) for minimally acceptable consensus.

The broad range of high-importance risk factors from both stages of this study cross multiple biopsychosocial domains and highlights five sectors of care that appear to impact upon readmission risk: pain and pain control, post-operative complications with or without the need for further surgery, need for supportive care post-operatively, comorbidities and their management, and surgeon-related factors. This reflects the difficulty and complexity of predicting 30-day readmission in this patient population [34] and the diverse range of factors that may increase the patient’s risk of being readmitted [35]. The findings of the thematic analysis supported this interpretation, with uncertainty and adaptability being the dominant themes emerging from the focus group discussion around risk factors for which votes were changed.

There was a lack of consensus on the importance of risk factors identified from systematic review and meta-analysis and on risk factors suggested by clinicians for which there was no evidence in prior literature. Coupled with the fact that a very broad variety of risk factors was put forward by clinicians, this suggested that the risk profile of a typical TKA patient readmitted within 30 days following readmission following TKA is not clear from a clinical perspective. Rather, there are many layers to hospital readmission following TKA. These include patient biopsychosocial factors, health system and hospital factors, and clinician-level factors, including decision-making processes regarding admission practices for patients who re-present to the hospital following their TKA surgery. As such, while clinicians agreed on a range of high-importance risk factors, there was a lack of consensus on the full range of diverse risk factors. This may be due to the fact that there is no biological mechanism clinicians can conceptualise to predict readmission. When predictor-outcome relationships can be conceptualised in this way, clinical insight may be more likely to reach a consensus because clinicians are trained to think in terms of pathophysiological mechanisms [36]. Clinicians may have a strong intuition based on their training and experience for a surgery-related outcome such as surgical site infection, which may, in turn, lead to readmission [37,38]. However, patients can be readmitted for many different reasons. Due to the complex range of factors that influence the decision to readmit a patient [39,40], different patients with very similar postoperative complications may have different decisions made regarding whether they should be readmitted

Perhaps clinicians would also be better equipped to identify risk factors for the most common organic indications for readmission, such as surgical site infection [38]. Future work could determine whether this approach could be taken to indirectly predict readmission by predicting its most common indications and mitigating these risks accordingly. This could start with a similar approach to the current study to obtain clinical insight into the importance of various risk factors for such complications.

### 4.1. Strengths

Strengths of this study include the recruitment of participants from two different specialities and levels of expertise and the selection of risk factors for appraisal being drawn from a robust systematic review and meta-analysis also carried about by the researchers who designed the current study. Participants were also recruited from a similar geographic region, facilitating appraisal of risk factors in terms of their relationship to local policies and practices and patterns of disease [41,42]. The use of an anonymous survey ensured participants felt free to vote based on their own appraisal rather than being influenced by the majority. The second stage of the study, in which risk factors without prior evidence in the literature were discussed, benefited from a structured discussion using elements of the nominal group technique. While consensus was poor in both stages of this study, the analytical approach built upon prior work on consensus among orthopaedic surgeons and anaesthetists around perioperative management and outcomes for TKA patients by providing a statistical measure of consensus not utilised in prior studies [23,43,44]. To the best of the authors’ knowledge, this is the first study in which a focus group with elements of the nominal group technique has been used for the purpose of evaluating risk factors for post-operative outcomes in orthopaedic surgery. Thematic analysis of the data generated from the focus group enabled us to gain a deeper understanding of the reasons why clinicians held certain views about the importance of various predictors [17,18]. This served to triangulate the survey and focus group findings and provide deeper insight into the complexities of identifying clinically relevant risk factors for 30-day readmission following TKA [17,45,46].

### 4.2. Limitations

Limitations included the low response rate and the fact that no orthopaedic residents, the most junior clinicians in the recruitment pool for this study, chose to participate. Despite this, the desired sample size was achieved for both stages of the study, with at least 30 for the Delphi survey [14] and between five and nine participants for the focus group [32]. However, self-selection bias could have influenced the findings, whereby the participants were a self-selected, more like-minded group of individuals [47]. Due to the need to keep the survey as brief as possible to facilitate the participation of busy clinicians, as well as concerns pertaining to protecting anonymity, the demographic information collected from participants was limited. Therefore, it is difficult to compare the demographic profile of the sample surveyed in this study with that of the national Australian orthopaedic and anaesthetic workforces [48,49] and subsequently determine whether respondents in this study differed from non-respondents. There could be differences between the respondents in this study and the broader clinician population, which could influence the generalizability of the findings [50].

Further complicating matters was the point made by one Delphi survey participant, a consultant anaesthetist, who pointed out that anaesthetists do not often find out about patient readmissions. While anaesthetists may have an impression of the patient’s likely post-operative course, this lack of ongoing follow-up makes predicting readmission particularly challenging.

### 4.3. Future Directions and Application of Findings

Future work could involve ascertaining the TKA caseload for Australian orthopaedic surgeons and anaesthetists and the length of time at a given designation (consultant, registrar, or resident for orthopaedic clinicians; consultant level for anaesthetists). This would facilitate a comparison of the sample surveyed in this study with the national average and therefore provide insight into the generalizability of findings.

Those risk factors identified as being highly important in both the Delphi and focus group phases of this study can be selected as variables in a risk prediction model for 30-day readmission for TKA patients [51,52,53,54]. The present study has highlighted the difficulty of predicting readmission due to the broad range of risk factors. This complexity may be difficult to reconcile from a clinical perspective, hence the poor consensus among clinicians, but the underlying complexity may be amenable to machine learning approaches that are capable of harnessing such intricacy in the data [55]. However, there is intrinsic value to involving clinicians in the model development process: (1) improving clinical interpretability [56]; (2) engaging stakeholders to enhance the clinical relevance of the model [53,57]; (3) inclusion of clinically meaningful risk factors into models that statistical predictor selection techniques may omit [15].

## 5. Conclusions

The broad range of high-importance risk factors from both stages of this study highlights five sectors of care that appear to impact upon readmission risk: pain and pain control, post-operative complications with or without the need for further surgery, need for supportive care post-operatively, comorbidities and their management, and surgeon-related factors. Thematic analysis of the focus group findings indicated that clinicians are faced with uncertainty and benefit from being adaptable in their approach to evaluating and mitigating readmission risk. High-importance risk factors in both the Delphi survey and focus group were identified. These risk factors were derived from a variety of clinical and biopsychosocial domains, including healthcare utilization, perioperative and intra-operative factors, clinician-related factors, comorbidity burden, socioeconomic status, and patient-reported outcome measures. Lack of consensus highlights the fact that this is a highly complex problem that is challenging to predict and which depends heavily on risk factors that may be open to interpretation, difficult to capture, and dependent upon personal clinical experience, which must be tailored to the individual patient.

## Figures and Tables

**Figure 1 jcm-12-00747-f001:**
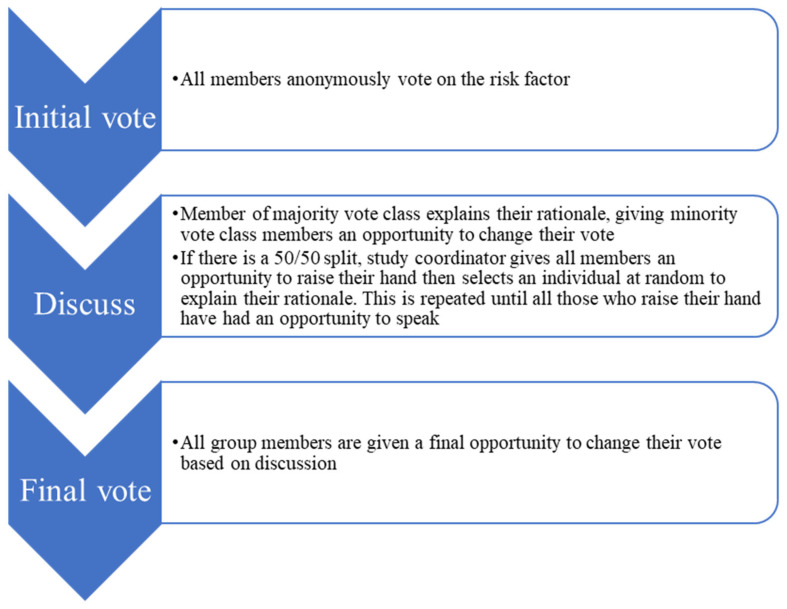
Focus group flow chart. This process is repeated for each risk factor included for discussion.

**Figure 2 jcm-12-00747-f002:**
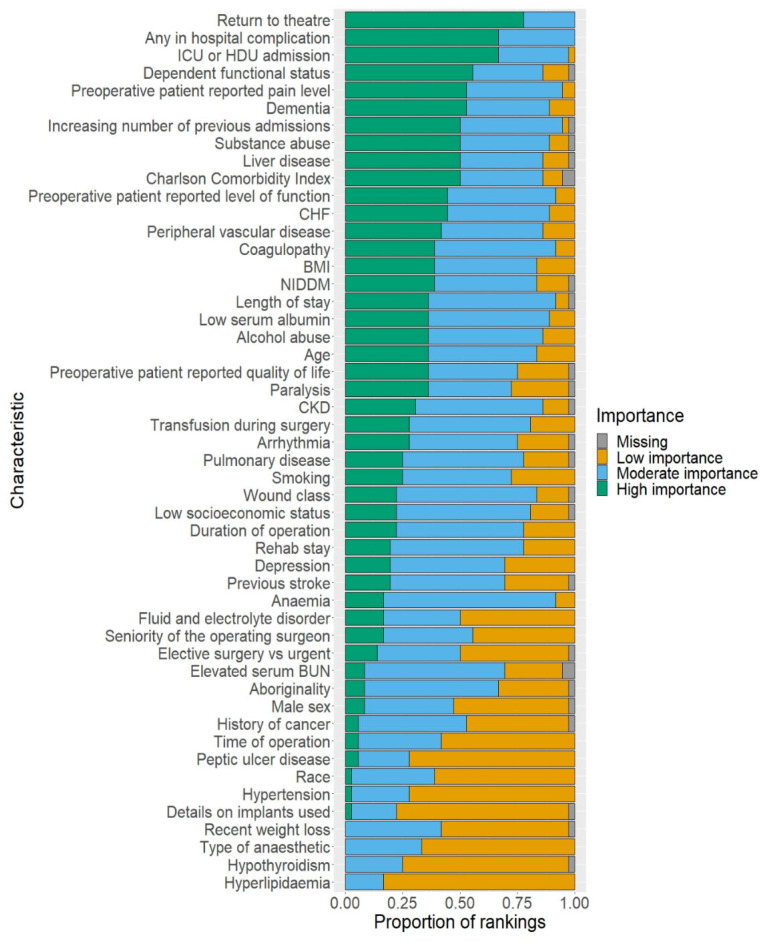
Counts of votes—Delphi survey. ICU = intensive care unit; HDU = high dependency unit; CHF = congestive heart failure; CKD = chronic kidney disease; NIDDM = non-insulin-dependent diabetes mellitus; BMI = body mass index.

**Figure 3 jcm-12-00747-f003:**
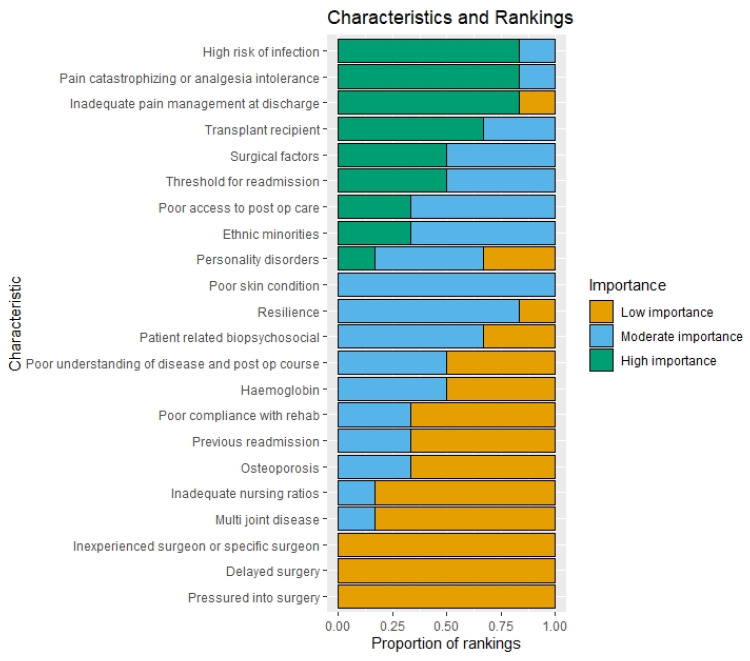
Counts of votes—focus group (final vote).

**Table 1 jcm-12-00747-t001:** Survey and focus group participant characteristics.

Characteristics	Delphi (*n* = 36) *	Focus Group (*n* = 6)
Designation (n)	CA = 20 CO = 13 Oreg = 3 Ores = 0	CA = 2 CO = 2 Oreg = 2 Ores = 0
Length of time at this level (years; mean (SD))	CA = 12.90 (7.44) CO = 20.23 (13.05) Oreg = 6.33 (1.15) Ores = N/A	CA = 12 (0.00) CO = 11 (9.90) Oreg = 4 (4.24) Ores = N/A
Caseload (number of TKAs per year; mean (SD))	CA = 26.6 (31.70) CO = 45.42 (33.94) ** Oreg = 72.50 (37.33) *** Ores = N/A	CA = 30 (0.00) CO = 70 (28.28) **** Oreg = 112.50 (53.03) ***** Ores = N/A

CA = consultant anaesthetist; CO = consultant orthopaedic surgeon; Oreg = orthopaedic registrar; Ores = orthopaedic resident; * 39 commenced the survey, but only 36 responded to any of the questions; ** there was one missing value, so the calculation was conducted on the non-missing values; *** one participant gave a range of 75–100 so a value of 87.5 was used in the calculation; **** one participant gave a range of 80–100 so a value of 90 was used in the calculation; ***** one participant gave a range of 50–100 so a value of 75 was used in the calculation.

**Table 2 jcm-12-00747-t002:** Delphi survey risk factors with a majority vote of high importance.

Groups:				
Overall (All Participants)	Consultant Anaesthetists	Consultant Orthopaedic Surgeons	Orthopaedic Registrars	All Orthopaedic Participants
Risk Factors:				
* Return to the theatre (0.75)	* Return to the theatre (0.70)	* Return to the theatre (0.77)	In-hospital complication any (1.00)	* Return to the theatre (0.81)
In-hospital complication any (0.67)	In-hospital complication any (0.65)	* ICU/HDU admission (0.69)	Dependent functional status (1.00)	* ICU/HDU admission (0.75)
* ICU/HDU admission (0.67)	* ICU/HDU admission (0.60)	Substance abuse (0.62)	Increasing number of previous admissions (1.00)	In-hospital complication any (0.69)
Dependent functional status (0.56)	Liver disease (0.55)	In-hospital complication any (0.62)	* Preoperative patient-reported pain level (1.00)	Charlson Comorbidity Index (0.56)
Dementia (0.53)	Congestive heart failure (0.55)	Liver disease (0.54)	* ICU/HDU admission (1.00)	Dependent functional status (0.56)
*Preoperative patient-reported pain level (0.53)	Dementia (0.55)	Charlson Comorbidity Index (0.54)	* Return to the theatre (1.00)	* Preoperative patient-reported pain level (0.56)
Liver disease (0.50)	Dependent functional status (0.55)		Charlson Comorbidity Index (0.67)	Substance abuse (0.50)
Charlson Comorbidity Index (0.50)	Increasing number of previous admissions (0.55)		Dementia (0.67)	Dementia (0.50)
Substance abuse (0.50)	Peripheral vascular disease (0.50)		* Preoperative patient-reported level of function (0.67)	
Increasing number of previous admissions (0.50)	Substance abuse (0.50)		* Duration of operation (0.67)	
Congestive heart failure (0.50)	* Preoperative patient-reported pain level (0.50)		Length of stay (0.67)	

ICU = intensive care unit; HDU = high dependency unit; * = majority (≥50%) voted as high-importance despite lack of systematic review evidence. Votes were provided overall and amongst each subgroup (numbers in brackets correspond to the proportion of votes in the high-importance category).

**Table 3 jcm-12-00747-t003:** Focus group risk factors with a final majority vote of high importance overall and amongst each subgroup (Number in brackets corresponds to proportion high importance vote).

Groups:				
Overall (All Participants)	Consultant Anaesthetists	Consultant Orthopaedic Surgeons	Orthopaedic Registrars	All Orthopaedic Participants
Risk Factors (Proportion of Votes in High Importance Category):				
Pain catastrophizing analgesia intolerance catastrophic pain (1.00)	Pain catastrophizing analgesia intolerance catastrophic pain (1.00)	Inadequate pain management at discharge (1.00)	High risk of infection immunocompromised state active IVDU infection in other primary joint replacement (1.00)	Pain catastrophizing analgesia intolerance catastrophic pain (1.00)
High risk of infection immunocompromised state active IVDU infection in other primary joint replacement (0.67)	High risk of infection immunocompromised state active IVDU infection in other primary joint replacement (0.50)	Surgical factors prolonged complex difficult surgery surgical misadventure (1.00)	Pain catastrophizing analgesia intolerance catastrophic pain (1.00)	High risk of infection immunocompromised state active IVDU infection in other primary joint replacement (0.75)
Transplant recipients better at self-managing medications and better screened for other comorbidities etc. (0.67)	Poor access to post-op care, lives far from a hospital, lack of access to allied health support, lack of access to telehealth support (0.50)	Pain catastrophizing analgesia intolerance catastrophic pain (1.00)	Inadequate pain management at discharge (0.50)	Inadequate pain management at discharge (0.75)
Inadequate pain management at discharge (0.50)	Surgical factors prolonged complex difficult surgery surgical misadventure (0.50)	Transplant recipients better at self-managing medications and better screened for other comorbidities etc. (1.00)	* Threshold for readmission, e.g., the specific ED or whether there is a junior registrar reviewing the patient (0.50)	Transplant recipients better at self-managing medications and better screened for other comorbidities etc. (0.75)
Surgical factors prolonged complex difficult surgery surgical misadventure (0.50)	Patient-related biopsychosocial lower education level, poor health literacy, non-English speaking (0.50)	High risk of infection immunocompromised state active IVDU infection in other primary joint replacement (0.50)	Transplant recipients better at self-managing medications and better screened for other comorbidities etc. (0.50)	Surgical factors prolonged complex difficult surgery surgical misadventure (0.50)
* Threshold for readmission, e.g., the specific ED or whether there is a junior registrar reviewing the patient (0.50)	Resilience (0.50)	Threshold for readmission, e.g., the specific ED or whether there is a junior registrar reviewing the patient (0.50)		* Threshold for readmission, e.g., the specific ED or whether there is a junior registrar reviewing the patient (0.50)
	* Threshold for readmission, e.g., the specific ED or whether there is a junior registrar reviewing the patient (0.50)	Personality disorders (0.50)		
	Transplant recipients better at self-managing medications and better screened for other comorbidities etc. (0.50)	Ethnic minorities—In the US: Black population; In Australia: Aboriginal and Torres Strait Islander Peoples, Refugees, Homelessness (0.50)		

* Moved into high importance category through discussion (therefore present here but absent from Appendix A); IVDU = intravenous drug use; ED = emergency department. Votes are provided overall and amongst each subgroup (numbers in brackets correspond to the proportion of votes in the high-importance category).

## Data Availability

Datasets supporting the conclusions of this article are included within the additional file. The vote numbers for the Delphi survey and focus group are available in Appendix A. The portion of the focus group transcript upon which the thematic analysis is based is available in Appendix A.

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
