# Peer review of "Established and Novel Risk Factors for 30-Day Readmission Following Total Knee Arthroplasty: A Modified Delphi and Focus Group Study to Identify Clinically Important Predictors"

_jcm, 2023, doi:10.3390/jcm12030747_

Round 1
Reviewer 1 Report
Thank you for the opportunity to review the paper.
Overall, I agree that readmissions following TKA are an important healt care issue, especially when taking into account that TKA rates are expected to rise.
The manuscript is well written and structured. However, the main concern is the response. Response rates of approximately 20% are extremely low and can cause a significant selction bias. What would the authors ecpect are the charactersitics of the 80% who did not respond. Are there any official characteristics to compare with? Please discuss more detailly.
Author Response
|
Comment |
Response |
Changes |
|
Thank you for the opportunity to review the paper.
Overall, I agree that readmissions following TKA are an important healt care issue, especially when taking into account that TKA rates are expected to rise.
The manuscript is well written and structured. However, the main concern is the response. Response rates of approximately 20% are extremely low and can cause a significant selction bias. What would the authors ecpect are the charactersitics of the 80% who did not respond. Are there any official characteristics to compare with? Please discuss more detailly. |
Thank you for your comment. This certainly was a challenge for the study and a very important consideration in the interpretation of findings, particularly regarding their generalisability. The following points have been added to the manuscript for more detailed discussion per your recommendation.
The response rates provided in the manuscript are a conservative estimate because site representatives at only five of the 11 eligible sites responded to the recruitment email. Therefore, while the estimated number of potentially eligible participants (i.e. the denominator) is provided for all 11 sites, the number of responses (i.e. the numerator) is possibly only from clinicians at the five sites whose site representatives responded to the recruitment email and confirmed they forwarded it to clinicians at their site. Therefore, the calculated response rate is likely an underestimate of the true response rate, with the latter likely being closer to prior published Delphi studies on prognostic factors (referenced in the main text).
While there are official statistics available on some demographic characteristics of orthopaedic surgeons and anaesthetists in Australia, comparing this broader population to the sample in this study is difficult because we collected minimal demographic information due to identifiability concerns and the preference to keep the survey as brief as possible to facilitate participation of busy clinicians. Future work could involve ascertaining the TKA caseload for Australian orthopaedic surgeons and anaesthetists, and length of time at given designation (consultant, registrar, or resident for orthopaedic clinicians; consultant level for anaesthetists). This would facilitate comparison of the sample surveyed in this study with the national average. |
The following was added to page 5, line 193: However, six of the 11 site representatives did not respond to the recruitment email, therefore the calculated response rates are likely an underestimate because the nu-merator (i.e. number of respondents) is possibly only provided for five of 11 sites, while the denominator represents the approximate number of respondents at all 11 sites. Therefore, the true response rate is likely somewhat closer to that of prior published Delphi survey studies on prognostic factors [8, 14].
The impact of this limitation was discussed in more detail in the ‘Limitations’ sub-section of the Discussion, with the following added to line 417 of page 13: Due to the need to keep the survey as brief as possible to facilitate the participation of busy clinicians, as well as concerns pertaining to protecting anonymity, the demo-graphic information collected from participants was limited. Therefore, it is difficult to compare the demographic profile of the sample surveyed in this study with that of the national Australian orthopaedic and anaesthetic workforces [48, 49] and subsequently determine whether respondents in this study differed from nonrespondents. There could be differences between the respondents in this study and the broader clinician population which could influence the generalizability of the findings [50].
Following the above addition to the ‘Limitations’ subsection of the Discussion, this statement was added to the ‘Future directions and application of findings’ subsection on page 13, line 432: Future work could involve ascertaining the TKA caseload for Australian orthopaedic surgeons and anaesthetists, and length of time at given desig-nation (consultant, registrar, or resident for orthopaedic clinicians; consultant level for anaesthetists). This would facilitate comparison of the sample surveyed in this study with the national average and therefore provide insight on the generalizability of findings. |
Reviewer 2 Report
Dear author,
I am pleased to submit to you my review of your article.
The topic is interesting, current, and relevant to our clinical practice.
The article is well written, but many concerns burden it with a minor revision before it can be accepted for publication.
Some suggested corrections have been listed below. Please answer point by point.
Minor revision in the manuscript:
GENERAL SECTION
-The manuscript is too long. Several general and well-known topics can be easily shortened without losing scientific information. The manuscript will improve with a reduction. It will be more straightforward, easier to read, and smoother. Remember that a manuscript should be as long as necessary but as short as possible.
-Use the third person throughout the text, and do not speak in the first person.
TITLE
-OK
ABSTRACT
-Well written.
INTRODUCTION
- I suggest adding a sentence at the end of this sentence that talks about recent innovations and trends that are bringing improvements in the clinical outcomes of TKA. I'll give you an example of a sentence you could insert: " In recent years, there has been an increase in new surgical techniques for TKA, in terms of alignment, prosthetic manufacture, and surgical procedure assistance with the Introduction of robotics in TKA. These have been introduced to improve clinical outcomes after TKA and promote immediate postoperative recovery by reducing the readmission rate." After the sentence, add these citations about the main innovations in this regard: doi: 10.1016/j.jor.2022.06.014; doi: 10.3390/s21165427; doi: 10.3390/app122111085 ; doi: 10.3390/jcm11216569.
MATERIALS AND METHODS
-Well written
RESULTS
-Tables 2 and 3: Uniform the bold in the header of the last column of both tables.
DISCUSSION
-Start the discussions by affirming the study's main finding, to be added at the beginning of the section.
CONCLUSION
-OK
REFERENCES
-Add recommended references. Check and correct.
Author Response
| Comment | Response | Changes |
|
General: The manuscript is too long. Several general and well-known topics can be easily shortened without losing scientific information. The manuscript will improve with a reduction. It will be more straightforward, easier to read, and smoother. Remember that a manuscript should be as long as necessary but as short as possible. |
Thank you for your comment. We agree brevity is crucial, while retaining clarity. In response, I have removed extraneous information from the manuscript in order to improve readability. |
Removed from introduction: The importance of reducing hospital readmission rates to improve quality of patient care and reduce healthcare expenditure has been recognised in such initiatives as the United States’ Hospital Readmissions Reduction Program [2]. In 2014, total knee arthroplasty (TKA) was added to this Program as a specific target for reducing readmission rates. Advanced osteoarthritis, the most common indication for TKA [3], is a growing problem [4]. There has been a concomitant increase in the annual rate of TKA utilisation and this increase is expected to continue [5, 6]. For TKA, readmission rates range between 3% and 6% on prior estimates [1, 7-9]. Given the continual increase in the annual rate of TKA utilisation, interventions targeted at reducing avoidable readmissions are a healthcare and financial priority.
Removed from Discussion (sub-section ‘Future directions and application of findings’): The Delphi technique has been used previously in clinical risk prediction model development studies [44-45]. This is in line with recommendations that the predictor selection process should in-corporate clinical expertise wherever possible, rather than relying solely on statistical techniques [46]. Accurate prediction of hospital readmissions for TKA patients is important for healthcare delivery – including shared decision-making, obtaining consent, patient health optimisation, discharge planning, and follow-up processes. Statistical and machine learning-based techniques can be utilised to build predictive models for this purpose [47]. Machine learning is a computational technique which harnesses statistical principles and iterative algorithms to identify and learn patterns in data from which predictors can be correlated with outcomes without requiring human specification of a set of rules or criteria [48]. The purpose of using computational and statistical principles is to optimise predictive accuracy. Such approaches utilising both clinical and statistical/machine learning expertise may increase clinical relevance [50] and statistical accuracy of the model [46]. |
|
General: Use the third person throughout the text, and do not speak in the first person. |
Thank you for bringing this to our attention. The manuscript has been thoroughly scanned for any such instances and corrected accordingly. |
Changes made throughout manuscript to remove ‘our’ and ‘we’: Page 2, line 62: - Original = First, we aimed… - Revised = The first aim was… Page 2, line 64: - Original = Second, we aimed… - Revised = The second aim was… Page 2, lines 76-77: - Original = We conducted the systematic review and meta-analysis on patient-related risk factors for 30-day readmission following TKA. - Revised (original sentence deleted, and the following added to the opening sentence of the ‘Study design’ section after “findings”) = on patient-related risk factors for 30-day readmission following TKA Page 4, line 145: - Original = We utilised… - Revised = Poll Everywhere was utilised… Page 5, line 170: - Original = …we aimed to recruit…for our target sample size - Revised = …the target sample size was… Page 5, line 185: - Original = …our interpretations… - Revised = …the interpretations… Page 11, line 344: - Original = …we engaged clinicians involved in the care of TKA patients… - Revised = …clinicians involved in the care of TKA patients were engaged… Page 11, line 354: - Original = …our thematic analysis… - Revised = …the thematic analysis… Page 12, line 397: - Original = …our knowledge… - Revised = …the authors’ knowledge… Page 12, line 407: - Original = …we tried to recruit in… - Revised = ….in the recruitment pool for… Page 12, line 408: - Original = …we achieved the desired sample size… - Revised = …the desired sample size was achieved for both stages of the study, with at least… |
|
Introduction: I suggest adding a sentence at the end of this sentence that talks about recent innovations and trends that are bringing improvements in the clinical outcomes of TKA. I'll give you an example of a sentence you could insert: " In recent years, there has been an increase in new surgical techniques for TKA, in terms of alignment, prosthetic manufacture, and surgical procedure assistance with the Introduction of robotics in TKA. These have been introduced to improve clinical outcomes after TKA and promote immediate postoperative recovery by reducing the readmission rate." After the sentence, add these citations about the main innovations in this regard: doi: 10.1016/j.jor.2022.06.014; doi: 10.3390/s21165427; doi: 10.3390/app122111085 ; doi: 10.3390/jcm11216569. |
Thank you for this recommendation. We have included these references in an explanatory sentence in the opening part of the Introduction. |
The second sentence of the Introduction now reads: In recent years, efforts to improve outcomes and reduce complications, including re-admission rate, following TKA have led to a surge in innovation in terms of alignment [2 - doi: 10.1016/j.jor.2022.06.014], prosthetic manufacture [3 - doi: 10.3390/jcm11216569], and assistance in the surgical procedure with robotics [4 - doi: 10.3390/app122111085] and other devices [5 - doi: 10.3390/s21165427]. |
|
Results: Tables 2 and 3: Uniform the bold in the header of the last column of both tables. |
Thank you for bringing this to our attention. The headers of these tables have been bolded as requested. |
Bolded all headers of Tables 2 and 3. |
|
Discussion: Start the discussions by affirming the study's main finding, to be added at the beginning of the section. |
Thank you for this recommendation. Immediately after summarising the aim of the study, we have added a summary of the main finding. |
Added the following immediately after the existing first sentence of the Discussion: Eleven risk factors received a majority (≥50%) vote of high importance in the Delphi survey overall: return to theatre, any in-hospital complication, intensive care unit or high-dependency unit admission, dependent functional status, dementia, preoperative patient-reported pain level, liver disease, Charlson Comorbidity Index, substance abuse, increasing number of previous admissions, and congestive heart failure. Six risk factors received a majority vote of high importance in the focus group overall: inadequate pain management at discharge, pain catastrophizing or analgesia intolerance/catastrophic pain, high risk of infection, transplant recipient, threshold for read-mission, and surgical factors (prolonged/difficult surgery). None of the Krippendorff’s alpha statistics reaching the threshold value (0.6) for minimally acceptable consensus. |
|
References: Add recommended references. Check and correct. |
Again, thank you for your recommendation. Per our previous response to your suggestion of an addition to the Introduction, these references have been checked and added. |
References added to the manuscript accordingly (see references 2 to 5 in the reference list). |